# Discriminative Transfer Learning with Tree-based Priors

**Nitish Srivastava**
Department of Computer Science
University of Toronto
nitish@cs.toronto.edu

**Ruslan Salakhutdinov**
Department of Computer Science and Statistics
University of Toronto
rsalakhu@cs.toronto.edu

## Abstract

High capacity classifiers, such as deep neural networks, often struggle on classes that have very few training examples. We propose a method for improving classification performance for such classes by discovering similar classes and transferring knowledge among them. Our method learns to organize the classes into a tree hierarchy. This tree structure imposes a prior over the classifier's parameters. We show that the performance of deep neural networks can be improved by applying these priors to the weights in the last layer. Our method combines the strength of discriminatively trained deep neural networks, which typically require large amounts of training data, with tree-based priors, making deep neural networks work well on infrequent classes as well. We also propose an algorithm for learning the underlying tree structure. Starting from an initial pre-specified tree, this algorithm modifies the tree to make it more pertinent to the task being solved, for example, removing semantic relationships in favour of visual ones for an image classification task. Our method achieves state-of-the-art classification results on the CIFAR-100 image data set and the MIR Flickr image-text data set.

## 1 Introduction

Learning classifiers that generalize well is a hard problem when only few training examples are available. For example, if we had only 5 images of a cheetah, it would be hard to train a classifier to be good at distinguishing cheetahs against hundreds of other classes, working off pixels alone. Any powerful enough machine learning model would severely overfit the few examples, unless it is held back by strong regularizers. This paper is based on the idea that performance can be improved using the natural structure inherent in the set of classes. For example, we know that cheetahs are related to tigers, lions, jaguars and leopards. Having labeled examples from these related classes should make the task of learning from 5 cheetah examples much easier. Knowing class structure should allow us to borrow "knowledge" from relevant classes so that only the distinctive features specific to cheetahs need to be learned. At the very least, the model should confuse cheetahs with these animals rather than with completely unrelated classes, such as cars or lamps. Our aim is to develop methods for transferring knowledge from related tasks towards learning a new task. In the endeavour to scale machine learning algorithms towards AI, it is imperative that we have good ways of transferring knowledge across related problems.

Finding relatedness is also a hard problem. This is because in the absence of any prior knowledge, in order to find which classes are related, we should first know what the classes are - i.e., have a good model for each one of them. But to learn a good model, we need to know which classes are related. This creates a cyclic dependency. One way to circumvent it is to use an external knowledge source, such as a human, to specify the class structure by hand. Another way to resolve this dependency is to iteratively learn a model of the what the classes are and what relationships exist between them, using one to improve the other. In this paper, we follow this bootstrapping approach.

This paper proposes a way of learning class structure and classifier parameters in the context of deep neural networks. The aim is to improve classification accuracy for classes with few examples. Deep neural networks trained discriminatively with back propagation achieved state-of-the-art performance on difficult classification problems with large amounts of labeled data [2, 14, 15]. The case of smaller amounts of data or datasets which contain rare classes has been relatively less studied. To address this shortcoming, our model augments neural networks with a tree-based prior over the last layer of weights. We structure the prior so that related classes share the same prior. This shared prior captures the features that are common across all members of any particular superclass. Therefore, a class with few examples, for which the model would otherwise be unable to learn good features for, can now have access to good features just by virtue of belonging to the superclass.

Learning a hierarchical structure over classes has been extensively studied in the machine learning, statistics, and vision communities. A large class of models based on hierarchical Bayesian models have been used for transfer learning [20, 4, 1, 3, 5]. The hierarchical topic model for image features of Bart et.al. [1] can discover visual taxonomies in an unsupervised fashion from large datasets but was not designed for rapid learning of new categories. Fei-Fei et.al. [5] also developed a hierarchical Bayesian model for visual categories, with a prior on the parameters of new categories that was induced from other categories. However, their approach is not well-suited as a generic approach to transfer learning because they learned a single prior shared across all categories. A number of models based on hierarchical Dirichlet processes have also been used for transfer learning [23, 17]. However, almost all of the the above-mentioned models are generative by nature. These models typically resort to MCMC approaches for inference, that are hard to scale to large datasets. Furthermore, they tend to perform worse than discriminative approaches, particularly as the number of labeled examples increases.

A large class of discriminative models [12, 25, 11] have also been used for transfer learning that enable discovering and sharing information among related classes. Most similar to our work is [18] which defined a generative prior over the classifier parameters and a prior over the tree structures to identify relevant categories. However, this work focused on a very specific object detection task and used an SVM model with pre-defined HOG features as its input. In this paper, we demonstrate our method on two different deep architectures (1) convolutional nets with pixels as input and single-label softmax outputs and (2) fully connected nets pretrained using deep Boltzmann machines with image features and text tokens as input and multi-label logistic outputs. Our model improves performance over strong baselines in both cases, lending some measure of universality to the approach. In essence, our model learns low-level features, high-level features, as well as a hierarchy over classes in an end-to-end way.

## 2 Model Description

Let $\mathcal{X} = \{\mathbf{x}^1, \mathbf{x}^2, \ldots, \mathbf{x}^N\}$ be a set of $N$ data points and $\mathcal{Y} = \{\mathbf{y}^1, \mathbf{y}^2, \ldots, \mathbf{y}^N\}$ be the set of corresponding labels, where each label $\mathbf{y}^i$ is a $K$ dimensional vector of targets. These targets could be binary, one-of-$K$, or real-valued. In our setting, it is useful to think of each $\mathbf{x}^i$ as an image and $\mathbf{y}^i$ as a one-of-$K$ encoding of the label. The model is a multi-layer neural network (see Fig. 1a). Let $\mathbf{w}$ denote the set of all parameters of this network (weights and biases for all the layers), excluding the top-level weights, which we denote separately as $\beta \in \mathbb{R}^{D \times K}$. Here $D$ represents the number of hidden units in the last hidden layer. The conditional distribution over $\mathcal{Y}$ can be expressed as

$$P(\mathcal{Y}|\mathcal{X}) = \int_{\mathbf{w},\beta} P(\mathcal{Y}|\mathcal{X}, \mathbf{w}, \beta) P(\mathbf{w}) P(\beta) d\mathbf{w} d\beta. \tag{1}$$

In general, this integral is intractable, and we typically resort to MAP estimation to determine the values of the model parameters $\mathbf{w}$ and $\beta$ that maximize

$$\log P(\mathcal{Y}|\mathcal{X}, \mathbf{w}, \beta) + \log P(\mathbf{w}) + \log P(\beta).$$

Here, $\log P(\mathcal{Y}|\mathcal{X}, \mathbf{w}, \beta)$ is the log-likelihood function and the other terms are priors over the model's parameters. A typical choice of prior is a Gaussian distribution with diagonal covariance:

$$\beta_k \sim \mathcal{N}\left(0, \frac{1}{\lambda} I_D\right), \forall k \in \{1, \ldots, K\}.$$

Here $\beta_k \in \mathbb{R}^D$ denotes the classifier parameters for class $k$. Note that this prior assumes that each $\beta_k$ is independent of all other $\beta_i$'s. In other words, a-priori, the weights for label $k$ are not related to any

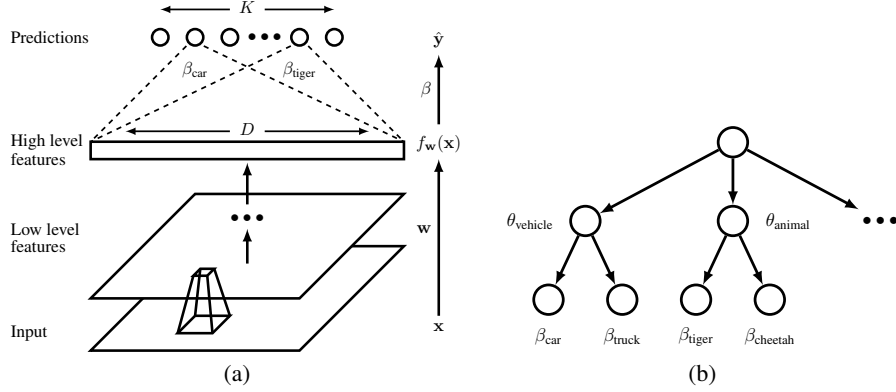

Figure 1: Our model: A deep neural network with priors over the classification parameters. The priors are derived from a hierarchy over classes.

other label's weights. This is a reasonable assumption when nothing is known about the labels. It works quite well for most applications with large number of labeled examples per class. However, if we know that the classes are related to one another, priors which respect these relationships may be more suitable. Such priors would be crucial for classes that only have a handful of training examples, since the effect of the prior would be more pronounced. In this work, we focus on developing such a prior.

## 2.1 Learning With a Fixed Tree Hierarchy

Let us first assume that the classes have been organized into a fixed tree hierarchy which is available to us. We will relax this assumption later by placing a hierarchical non-parametric prior over the tree structures. For ease of exposition, consider a two-level hierarchy[1], as shown in Fig. 1b. There are $K$ leaf nodes corresponding to the $K$ classes. They are connected to $S$ super-classes which group together similar basic-level classes. Each leaf node $k$ is associated with a weight vector $\beta_k \in \mathbb{R}^D$. Each super-class node $s$ is associated with a vector $\theta_s \in \mathbb{R}^D$, $s = 1, ..., S$. We define the following generative model for $\beta$

$$\theta_s \sim \mathcal{N}\left(0, \frac{1}{\lambda_1}I_D\right), \qquad \beta_k \sim \mathcal{N}\left(\theta_{\text{parent}(k)}, \frac{1}{\lambda_2}I_D\right). \tag{2}$$

This prior expresses relationships between classes. For example, it asserts that $\beta_{\text{car}}$ and $\beta_{\text{truck}}$ are both deviations from $\theta_{\text{vehicle}}$. Similarly, $\beta_{\text{cat}}$ and $\beta_{\text{dog}}$ are deviations from $\theta_{\text{animal}}$. Eq. 1 can now be re-written to include $\theta$ as follows

$$P(\mathcal{Y}|\mathcal{X}) = \int_{\mathbf{w},\beta,\theta} P(\mathcal{Y}|\mathcal{X}, \mathbf{w}, \beta)P(\mathbf{w})P(\beta|\theta)P(\theta)d\mathbf{w}d\beta d\theta. \tag{3}$$

We can perform MAP inference to determine the values of $\{\mathbf{w}, \beta, \theta\}$ that maximize

$$\log P(\mathcal{Y}|\mathcal{X}, \mathbf{w}, \beta) + \log P(\mathbf{w}) + \log P(\beta|\theta) + \log P(\theta).$$

In terms of a loss function, we wish to minimize

$$
\begin{aligned}
L(\mathbf{w}, \beta, \theta) &= -\log P(\mathcal{Y}|\mathcal{X}, \mathbf{w}, \beta) - \log P(\mathbf{w}) - \log P(\beta|\theta) - \log P(\theta) \\
&= -\log P(\mathcal{Y}|\mathcal{X}, \mathbf{w}, \beta) + \frac{\lambda^2}{2}||\mathbf{w}||^2 + \frac{\lambda_2}{2}\sum_{k=1}^{K}||\beta_k - \theta_{\text{parent}(k)}||^2 + \frac{\lambda_1}{2}||\theta||^2. \quad (4)
\end{aligned}
$$

Note that by fixing the value of $\theta = 0$, this loss function recovers our standard loss function. The choice of normal distributions in Eq. 2 leads to a nice property that maximization over $\theta$, given $\beta$ can be done in closed form. It just amounts to taking a (scaled) average of all $\beta_k$'s which are children of $\theta_s$. Let $C_s = \{k|\text{parent}(k) = s\}$, then

$$\theta_s^* = \frac{1}{|C_s| + \lambda_1/\lambda_2}\sum_{k \in C_s}.\beta_k \tag{5}$$

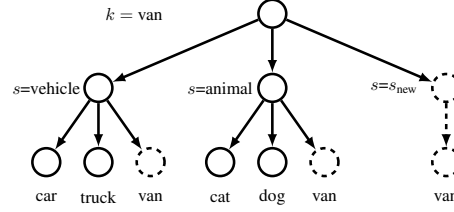

```
 1: Given: 𝒳, 𝒴, classes K, superclasses S, initial z, L, M.
 2: Initialize w, β.
 3: repeat
 4:     // Optimize w, β with fixed z.
 5:     w, β ← SGD (𝒳, 𝒴, w, β, z) for L steps.
 6:     // Optimize z, β with fixed w.
 7:     RandomPermute(K)
 8:     for k in K do
 9:        for s in S ∪ {s_new} do
10:           z_k ← s
11:           β^s ← SGD (f_w(𝒳), 𝒴, β, z) for M steps.
12:        end for
13:        s' ← ChooseBestSuperclass(β^1, β^2, …)
14:        β ← β^{s'}, z_k ← s', S ← S ∪ {s'}
15:     end for
16: until convergence
```

Algorithm 1: Procedure for learning the tree.

Therefore, the loss function in Eq. 4 can be optimized by iteratively performing the following two steps. In the first step, we maximize over $\mathbf{w}$ and $\beta$ keeping $\theta$ fixed. This can be done using standard stochastic gradient descent (SGD). Then, we maximize over $\theta$ keeping $\beta$ fixed. This can be done in closed form using Eq. 5. In practical terms, the second step is almost instantaneous and only needs to be performed after every $T$ gradient descent steps, where $T$ is around 10-100. Therefore, learning is almost identical to standard gradient descent. It allows us to exploit the structure over labels at a very nominal cost in terms of computational time.

## 2.2 Learning the Tree Hierarchy

So far we have assumed that our model is given a fixed tree hierarchy. Now, we show how the tree structure can be learned during training. Let $\mathbf{z}$ be a $K$-length vector that specifies the tree structure, that is, $z_k = s$ indicates that class $k$ is a child of super-class $s$. We place a non-parametric Chinese Restaurant Process (CRP) prior over $\mathbf{z}$. This prior $P(\mathbf{z})$ gives the model the flexibility to have any number of superclasses. The CRP prior extends a partition of $k$ classes to a new class by adding the new class either to one of the existing superclasses or to a new superclass. The probability of adding it to superclass $s$ is $\frac{c^s}{k+\gamma}$ where $c^s$ is the number of children of superclass $s$. The probability of creating a new superclass is $\frac{\gamma}{k+\gamma}$. In essence, it prefers to add a new node to an existing large superclass instead of spawning a new one. The strength of this preference is controlled by $\gamma$.

Equipped with the CRP prior over $\mathbf{z}$, the conditional over $\mathcal{Y}$ takes the following form

$$P(\mathcal{Y}|\mathcal{X}) = \sum_{\mathbf{z}} \left( \int_{\mathbf{w},\beta,\theta} P(\mathcal{Y}|\mathcal{X}, \mathbf{w}, \beta) P(\mathbf{w}) P(\beta|\theta, \mathbf{z}) P(\theta) d\mathbf{w} d\beta d\theta \right) P(\mathbf{z}). \tag{6}$$

MAP inference in this model leads to the following optimization problem

$$\max_{\mathbf{w},\beta,\theta,\mathbf{z}} \log P(\mathcal{Y}|\mathcal{X}, \mathbf{w}, \beta) + \log P(\mathbf{w}) + \log P(\beta|\theta, \mathbf{z}) + \log P(\theta) + \log P(\mathbf{z}).$$

Maximization over $\mathbf{z}$ is problematic because the domain of $\mathbf{z}$ is a huge discrete set. Fortunately, this can be approximated using a simple and parallelizable search procedure.

We first initialize the tree sensibly. This can be done by hand or by extracting a semantic tree from WordNet [16]. Let the number of superclasses in the tree be $S$. We optimize over $\{\mathbf{w}, \beta, \theta\}$ for a $L$ steps using this tree. Then, a leaf node is picked uniformly at random from the tree and $S + 1$ tree proposals are generated as follows. $S$ proposals are generated by attaching this leaf node to each of the $S$ superclasses. One additional proposal is generated by creating a new super-class and attaching the label to it. This process is shown in Algorithm 1. We then re-estimate $\{\beta, \theta\}$ for each of these $S + 1$ trees for a few steps. Note that each of the $S + 1$ optimization problems can be performed independently, in parallel. The best tree is then picked using a validation set. This process is repeated by picking another node and again trying all possible locations for it. After each node has been picked once and potentially repositioned, we take the resulting tree and go back to

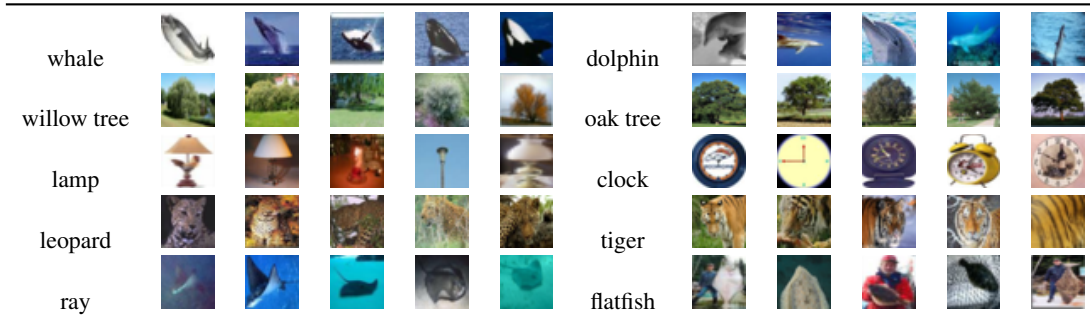

Figure 2: Examples from CIFAR-100. Five randomly chosen examples from 8 of the 100 classes are shown. Classes in each row belong to the same superclass.

optimizing $\mathbf{w}, \beta$ using this newly learned tree in place of the given tree. If the position of any class in the tree did not change during a full pass through all the classes, the hierarchy discovery was said to have converged. When training this model on CIFAR-100, this amounts to interrupting the stochastic gradient descent after every 10,000 steps to find a better tree. The amount of time spent in learning this tree is a small fraction of the total time (about 5%).

## 3 Experiments on CIFAR-100

The CIFAR-100 dataset [13] consists of $32 \times 32$ color images belonging to 100 classes. These classes are divided into 20 groups of 5 each. For example, the superclass **fish** contains *aquarium fish*, *flatfish*, *ray*, *shark* and *trout*; and superclass **flowers** contains *orchids*, *poppies*, *roses*, *sunflowers* and *tulips*. Some examples from this dataset are shown in Fig. 2. We chose this dataset because it has a large number of classes with a few examples in each, making it ideal for demonstrating the utility of transfer learning. There are only 600 examples of each class of which 500 are in the training set and 100 in the test set. We preprocessed the images by doing global contrast normalization followed by ZCA whitening.

### 3.1 Model Architecture and Training Details

We used a convolutional neural network with 3 convolutional hidden layers followed by 2 fully connected hidden layers. All hidden units used a rectified linear activation function. Each convolutional layer was followed by a max-pooling layer. Dropout [8] was applied to all the layers of the network with the probability of retaining a hidden unit being $p = (0.9, 0.75, 0.75, 0.5, 0.5, 0.5)$ for the different layers of the network (going from input to convolutional layers to fully connected layers). Max-norm regularization [8] was used for weights in both convolutional and fully connected layers. The initial tree was chosen based on the superclass structure given in the data set. We learned a tree using Algorithm 1 with $L = 10,000$ and $M = 100$. The final learned tree is provided in the supplementary material.

### 3.2 Experiments with Few Examples per Class

In our first set of experiments, we worked in a scenario where each class has very few examples. The aim was to assess whether the proposed model allows related classes to borrow information from each other. For a baseline, we used a standard convolutional neural network with the same architecture as our model. This is an extremely strong baseline and already achieved excellent results, outperforming all previously reported results on this dataset as shown in Table 1. We created 5 subsets of the data by randomly choosing 10, 25, 50, 100 and 250 examples per class, and trained four models on each subset. The first was the baseline. The second was our model using the given tree structure (100 classes grouped into 20 superclasses) which was kept fixed during training. The third and fourth were our models with a learned tree structure. The third one was initialized with a random tree and the fourth with the given tree. The random tree was constructed by drawing a sample from the CRP prior and randomly assigning classes to leaf nodes.

The test performance of these models is compared in Fig. 3a. We observe that if the number of examples per class is small, the fixed tree model already provides significant improvement over the baseline. The improvement diminishes as the number of examples increases and eventually the performance falls below the baseline (61.7% vs 62.8%). However, the learned tree model does

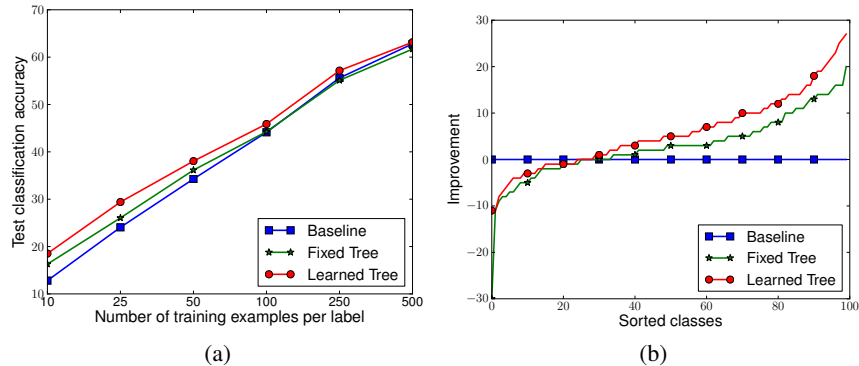

|     | (a) |     | (b) |
|-----|-----|-----|-----|

Figure 3: Classification results on CIFAR-100. **Left**: Test set classification accuracy for different number of training examples per class. **Right**: Improvement over the baseline when trained on 10 examples per class. The learned tree models were initialized at the given tree.

| Method | Test Accuracy % |
|--------|-----------------|
| Conv Net + max pooling | $56.62 \pm 0.03$ |
| Conv Net + stochastic pooling [24] | 57.49 |
| Conv Net + maxout [6] | 61.43 |
| Conv Net + max pooling + dropout (Baseline) | $62.80 \pm 0.08$ |
| Baseline + fixed tree | $61.70 \pm 0.06$ |
| Baseline + learned tree (Initialized randomly) | $61.20 \pm 0.35$ |
| Baseline + learned tree (Initialized from given tree) | $\mathbf{63.15 \pm 0.15}$ |

Table 1: Classification results on CIFAR-100. All models were trained on the full training set.

better. Even with 10 examples per class, it gets an accuracy of 18.52% compared to the baseline model's 12.81% or the fixed tree model's 16.29%. Thus the model can get almost a 50% relative improvement when few examples are available. As the number of examples increases, the relative improvement decreases. However, even for 500 examples per class, the learned tree model improves upon the baseline, achieving a classification accuracy of 63.15%. Note that initializing the model with a random tree decreases model performance, as shown in Table 1.

Next, we analyzed the learned tree model to find the source of the improvements. We took the model trained on 10 examples per class and looked at the classification accuracy separately for each class. The aim was to find which classes gain or suffer the most. Fig. 3b shows the improvement obtained by different classes over the baseline, where the classes are sorted by the value of the improvement over the baseline. Observe that about 70 classes benefit in different degrees from learning a hierarchy for parameter sharing, whereas about 30 classes perform worse as a result of transfer. For the learned tree model, the classes which improve most are *willow tree* (+26%) and *orchid* (+25%). The classes which lose most from the transfer are *ray* (-10%) and *lamp* (-10%).

We hypothesize that the reason why certain classes gain a lot is that they are very similar to other classes within their superclass and thus stand to gain a lot by transferring knowledge. For example, the superclass for *willow tree* contains other trees, such as *maple tree* and *oak tree*. However, *ray* belongs to superclass *fish* which contains more typical examples of fish that are very dissimilar in appearance. With the fixed tree, such transfer hurts performance (*ray* did worse by -29%). However, when the tree was learned, this class split away from the *fish* superclass to join a new superclass and did not suffer as much. Similarly, *lamp* was under *household electrical devices* along with *keyboard* and *clock*. Putting different kinds of electrical devices under one superclass makes semantic sense but does not help for visual recognition tasks. This highlights a key limitation of hierarchies based on semantic knowledge and advocates the need to learn the hierarchy so that it becomes relevant to the task at hand. The full learned tree is provided in the supplementary material.

### 3.3 Experiments with Few Examples for One Class

In this set of experiments, we worked in a scenario where there are lots of examples for different classes, but only few examples of one particular class. The aim was to see whether the model transfers information from other classes that it has learned to this "rare" class. We constructed training sets by randomly drawing either 5, 10, 25, 50, 100, 250 or 500 examples from the *dolphin*

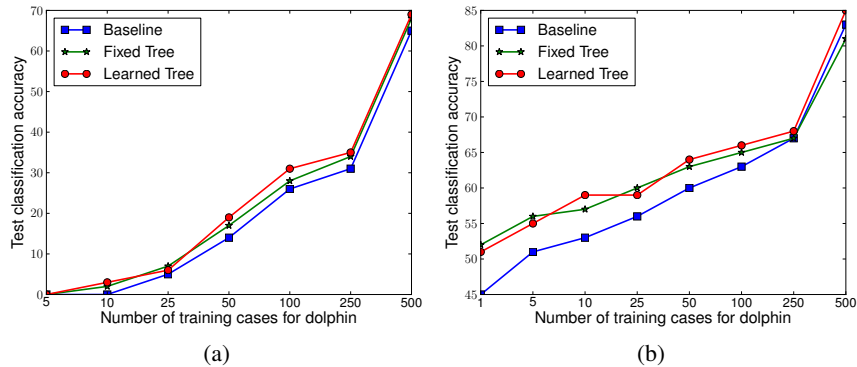

|  | (a) | | (b) |

Figure 4: Results on CIFAR-100 with few examples for the *dolphin* class. **Left**: Test set classification accuracy for different number of examples. **Right**: Accuracy when classifying a *dolphin* as *whale* or *shark* is also considered correct.

| **Classes** | baby, female, people, portrait | plant life, river, water | clouds, sea, sky, transport, water | animals, dog, food | clouds, sky, structures |
|---|---|---|---|---|---|
| **Images** | 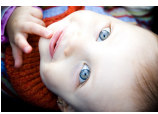 |  |  |  |  |
| **Tags** | claudia | ⟨ no text ⟩ | barco, pesca, boattosail, navegação | watermelon, dog, hilarious, chihuahua | colors, cores, centro, commercial, building |

Figure 5: Some examples from the MIR-Flickr dataset. Each instance in the dataset is an image along with textual tags. Each image has multiple classes.

class and all 500 training examples for the other 99 classes. We trained the baseline, fixed tree and learned tree models with each of these datasets. The objective was kept the same as before and no special attention was paid to the *dolphin* class. Fig. 4a shows the test accuracy for correctly predicting the *dolphin* class. We see that transfer learning helped tremendously. For example, with 10 cases, the baseline gets 0% accuracy whereas the transfer learning model can get around 3%. Even for 250 cases, the learned tree model gives significant improvements (31% to 34%). We repeated this experiment for classes other than *dolphin* as well and found similar improvements. See the supplementary material for a more detailed description.

In addition to performing well on the class with few examples, we would also want any errors to be sensible. To check if this was indeed the case, we evaluated the performance of the above models treating the classification of *dolphin* as *shark* or *whale* to also be correct, since we believe these to be reasonable mistakes. Fig. 4b shows the classification accuracy under this assumption for different models. Observe that the transfer learning methods provide significant improvements over the baseline. Even when we have just 1 example for *dolphin*, the accuracy jumps from 45% to 52%.

## 4 Experiments on MIR Flickr

The Multimedia Information Retrieval Flickr Data set [9] consists of 1 million images collected from the social photography website Flickr along with their user assigned tags. Among the 1 million images, 25,000 have been annotated using 38 labels. These labels include object categories such as, *bird*, *tree*, *people*, as well as scene categories, such as *indoor*, *sky* and *night*. Each image has multiple labels. Some examples are shown in Fig. 5.

This dataset is different from CIFAR-100 in many ways. In the CIFAR-100 dataset, our model was trained using image pixels as input and each image belonged to only one class. MIR-FLickr is a multimodal dataset for which we used standard computer vision image features and word counts as inputs. The CIFAR-100 models used a multi-layer convolutional network, whereas for this dataset we use a fully connected neural network initialized by unrolling a Deep Boltzmann Machine (DBM) [19]. Moreover, this dataset offers a more natural class distribution where some classes occur more often than others. For example, *sky* occurs in over 30% of the instances, whereas *baby* occurs in fewer than 0.4%. We also used 975,000 unlabeled images for unsupervised training of the DBM. We use the publicly available features and train-test splits from [21].

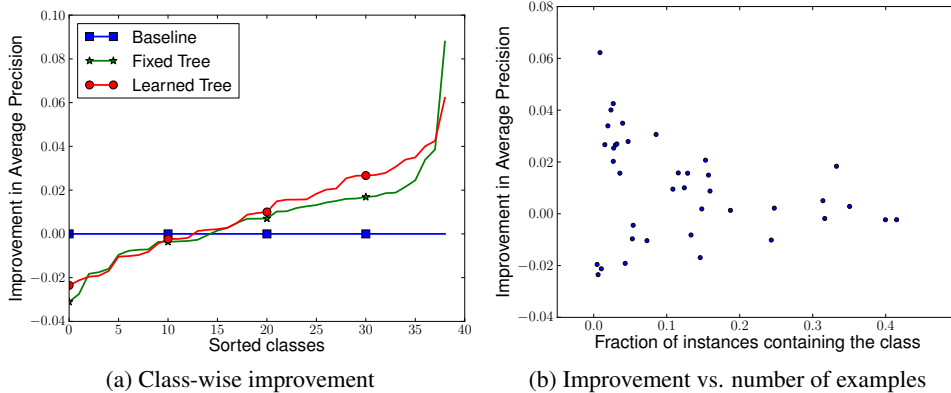

| (a) Class-wise improvement | (b) Improvement vs. number of examples |

Figure 6: Results on MIR Flickr. **Left**: Improvement in Average Precision over the baseline for different methods. **Right**: Improvement of the learned tree model over the baseline for different classes along with the fraction of test cases which contain that class. Each dot corresponds to a class. Classes with few examples (towards the left of plot) usually get significant improvements.

| Method | MAP |
|---|---|
| Logistic regression on Multimodal DBM [21] | 0.609 |
| Multiple Kernel Learning SVMs [7] | 0.623 |
| TagProp [22] | 0.640 |
| Multimodal DBM + finetuning + dropout (Baseline) | $0.641 \pm 0.004$ |
| Baseline + fixed tree | $0.648 \pm 0.004$ |
| Baseline + learned tree (initialized from given tree) | $\mathbf{0.651 \pm 0.005}$ |

Table 2: Mean Average Precision obtained by different models on the MIR-Flickr data set.

## 4.1 Model Architecture and Training Details

In order to make our results directly comparable to [21], we used the same network architecture as described therein. The authors of the dataset [10] provided a high-level categorization of the classes which we use to create an initial tree. This tree structure and the one learned by our model are shown in the supplementary material. We used Algorithm 1 with $L = 500$ and $M = 100$.

## 4.2 Classification Results

For a baseline we used a Multimodal DBM model after finetuning it discriminatively with dropout. This model already achieves state-of-the-art results, making it a very strong baseline. The results of the experiment are summarized in Table 2. The baseline achieved a MAP of 0.641, whereas our model with a fixed tree improved this to 0.647. Learning the tree structure further pushed this up to 0.651. For this dataset, the learned tree was not significantly different from the given tree. Therefore, we expected the improvement from learning the tree to be marginal. However, the improvement over the baseline was significant, showing that transferring information between related classes helped.

Looking closely at the source of gains, we found that similar to CIFAR-100, some classes gain and others lose as shown in Fig. 6a. It is encouraging to note that classes which occur rarely in the dataset improve the most. This can be seen in Fig. 6b which plots the improvements of the learned tree model over the baseline against the fraction of test instances that contain that class. For example, the average precision for *baby* which occurs in only 0.4% of the test cases improves from 0.173 (baseline) to 0.205 (learned tree). This class borrows from *people* and *portrait* both of which occur very frequently. The performance on *sky* which occurs in 31% of the test cases stays the same.

## 5 Conclusion

We proposed a model that augments standard neural networks with tree-based priors over the classification parameters. These priors follow the hierarchical structure over classes and enable the model to transfer knowledge from related classes. We also proposed a way of learning the hierarchical structure. Experiments show that the model achieves excellent results on two challenging datasets.

## Footnotes

[1]The model can be easily generalized to deeper hierarchies.

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
