[Supplementary Material · supplementary_material.pdf]

# Supplementary Material for Discriminative Transfer Learning with Tree-based Priors

**Nitish Srivastava**
Department of Computer Science
University of Toronto
nitish@cs.toronto.edu

**Ruslan Salakhutdinov**
Department of Computer Science and Statistics
University of Toronto
rsalakhu@cs.toronto.edu

## 1   Experimental Details for CIFAR-100

In this section we describe the experimental setup for CIFAR-100, including details of the architecture and the structures of the initial and learned trees.

### 1.1   Architecture and training details

We choose the architecture that worked best for the baseline model using a validation set. All the experiments reported in the paper for this dataset use this network architecture. It consists of three convolutional layers followed by 2 fully connected layers. Each convolutional layer is followed by a max-pooling layer. The convolutional layers have 96, 128 and 256 filters respectively. Each convolutional layer has a $5 \times 5$ receptive field applied with a stride of 1 pixel. Each max pooling layer pools $3 \times 3$ regions at strides of 2 pixels. The two fully connected hidden layers having 2048 units each. All units use the rectified linear activation function. Dropout was applied to all the layers of the network with the probability of retaining the unit being $p = (0.9, 0.75, 0.75, 0.5, 0.5, 0.5)$ for the different layers of the network (going from input to convolutional layers to fully connected layers). In addition, the max-norm constraint with $c = 4$ was used for all the weights.

Training the model on very small subsets of the data poses multiple problems. One of them is holding out a validation set. When training with 10 examples of class, we want to mimic a situation when we just have these 10 examples, and not a separate large validation set which has many more examples of this class. Therefore, we held out 30% of training data as validation, even when only 10 training examples are present. In order to remove noise, we repeated the experiment multiple times, selecting 7 random training cases and 3 random validation cases. Once we used the validation set to determine the hyperparameters, we combined it with the training set and trained the model down to the training set cross entropy that was obtained when the validation set was kept separate. This allows us to use the full training set which is crucial in very small dataset regimes.

### 1.2   Initial and Learned Trees

The authors of the CIFAR-100 data set [2] divide the set of 100 classes into 20 superclasses. These classes are shown in Table 1. Using this tree as an initialization, we used our model to learn the tree. The learned tree is shown in Table 2. This tree was learned with 50 examples per class. The superclasses do not have names since they were learned. We can see that the tree makes some sensible moves. For example, it creates a superclass for *shark*, *dolphin* and *whale*. It creates a superclass for worm-like creatures such as *caterpillar*, *worm* and *snake*. However this class also includes *snail* which is harder to explain. It includes *television* along with other furniture items, removing it from the house hold electrical devices superclass. However, some choices do not seem good. For example, *clock* is put together with *bicycle* and *motorcycle*. The only similarity between them is presence of circular objects.

| Superclass | Classes |
|---|---|
| aquatic mammals | dolphin, whale, seal, otter, beaver |
| fish | aquarium fish, flatfish, ray, shark, trout |
| flowers | orchid, poppy, rose, sunflower, tulip |
| food containers | bottle, bowl, can, cup, plate |
| fruit and vegetables | apple, mushroom, orange, pear, sweet pepper |
| household electrical devices | clock, keyboard, lamp, telephone, television |
| household furniture | bed, chair, couch, table, wardrobe |
| insects | bee, beetle, butterfly, caterpillar, cockroach |
| large carnivores | bear, leopard, lion, tiger, wolf |
| large man made outdoor things | bridge, castle, house, road, skyscraper |
| large natural outdoor scenes | cloud, forest, mountain, plain, sea |
| large omnivores and herbivores | camel, cattle, chimpanzee, elephant, kangaroo |
| medium sized mammals | fox, porcupine, possum, raccoon, skunk |
| non insect invertebrates | crab, lobster, snail, spider, worm |
| people | baby, boy, girl, man, woman |
| reptiles | crocodile, dinosaur, lizard, snake, turtle |
| small mammals | hamster, mouse, rabbit, shrew, squirrel |
| trees | maple tree, oak tree, palm tree, pine tree, willow tree |
| vehicles 1 | bicycle, bus, motorcycle, pickup truck, train |
| vehicles 2 | lawn mower, rocket, streetcar, tank, tractor |

Table 1: Fixed tree hierarchy for the CIFAR-100 dataset.

| Superclass | Classes |
|---|---|
| superclass 1 | dolphin, whale, shark |
| superclass 2 | aquarium fish, trout, flatfish |
| superclass 3 | orchid, poppy, rose, sunflower, tulip, butterfly |
| superclass 4 | bottle, bowl, can, cup, plate |
| superclass 5 | apple, mushroom, orange, pear, sweet pepper |
| superclass 6 | keyboard, telephone |
| superclass 7 | bed, chair, couch, table, wardrobe, television |
| superclass 8 | bee, beetle, cockroach, lobster |
| superclass 9 | bear, leopard, lion, tiger, wolf |
| superclass 10 | castle, house, skyscraper, train |
| superclass 11 | cloud, forest, mountain, plain, sea |
| superclass 12 | camel, cattle, chimpanzee, elephant, kangaroo, dinosaur |
| superclass 13 | fox, porcupine, possum, raccoon, skunk |
| superclass 14 | snail, worm, snake, caterpillar, ray |
| superclass 15 | baby, boy, girl, man, woman |
| superclass 16 | crocodile, lizard, turtle |
| superclass 17 | hamster, mouse, rabbit, shrew, squirrel, beaver, otter |
| superclass 18 | maple tree, oak tree, palm tree, pine tree, willow tree |
| superclass 19 | streetcar, bus, motorcycle, road |
| superclass 20 | lawn mower, pickup truck, tank, tractor |
| superclass 21 | bicycle, motorcycle, clock |
| superclass 22 | crab, spider |
| superclass 23 | bridge, seal |
| superclass 24 | rocket |
| superclass 25 | lamp |

Table 2: Learned tree hierarchy for the CIFAR-100 dataset.

# 2   Experimental Details for MIR Flickr

In this section we describe the experimental setup for MIR Flickr, including details of the architecture and the structures of the initial and learned trees.

## 2.1 Architecture and training details

The model was initialized by unrolling a multimodal DBM. The DBM consisted of two pathways. The image pathway had 3857 input units, followed by 2 hidden layers of 1024 hidden units. The text pathway had 2000 input units (corresponding the top 2000 most frequent tags in the dataset), followed by 2 hidden layers of 1024 hidden units each. These two pathways were combined at the joint layer which had 2048 hidden units. All hidden units were logistic. The DBM was trained using publicly available features and code. We additionally finetuned the entire model using dropout and used that as our baseline. During dropout, each unit was retained with probability $p = 0.8$ at each layer. We split the 25,000 labeled examples into 10,000 for training, 5,000 for validation and 10,000 for test.

## 2.2 Initial and Learned Trees

This dataset contains 38 classes, which can be categorized under a tree as shown in Table 3. The superclasses are based on the higher-level concepts as specified in [1]. The learned tree is shown in Table 4.

| Superclass | Classes |
|---|---|
| animals | animals, bird, dog, bird*, dog* |
| people | baby*, people*, female*, male*, portrait*, baby, people, female, male, portrait |
| plants | flower, tree, plant life, flower*, tree* |
| sky | clouds, clouds*, sky |
| water | water, river, lake, sea, river*, sea* |
| time of day | night, night*, sunset |
| transport | transport, car, car* |
| structures | structures |
| food | food |
| indoor | indoor |

Table 3: Given tree hierarchy for the MIR-Flickr dataset.

| Superclass | Classes |
|---|---|
| superclass 1 | animals, dog, dog*, bird* |
| superclass 2 | bird, sky, clouds, clouds* |
| superclass 3 | baby*, baby, people*, female*, male*, portrait*, portrait |
| superclass 4 | people, female, male |
| superclass 5 | flower, flower*, tree* |
| superclass 6 | tree, plant life |
| superclass 7 | water, river, sea |
| superclass 8 | lake, river*, sea* |
| superclass 9 | night, night* |
| superclass 10 | transport |
| superclass 11 | car, car* |
| superclass 12 | sunset |
| superclass 13 | structures |
| superclass 14 | food |
| superclass 15 | indoor |

Table 4: Learned tree hierarchy for the MIR-Flickr dataset.

## 3 Additional experiments on CIFAR-100 with few examples for one class

In this section we describe additional experiments in which we worked in a scenario where there are lots of examples for different classes, but only few examples of one particular class. In the paper, we presented details for the *dolphin* class. Here we include details of the same experiment with two other classes. The class presented in the paper is a typical case. Here, we pick two classes – one on which the model does quite well and the other on which it performs poorly. We take the *orchid* class

Figure 1: Results on CIFAR-100 with few examples for the *orchid* class. **Left**: Test set classification accuracy for different number of examples. **Right**: Accuracy when classifying a *orchid* as any other kind of flower is also considered correct.

Figure 2: Results on CIFAR-100 with few examples for the *lamp* class. **Left**: Test set classification accuracy for different number of examples. **Right**: Accuracy when classifying a *lamp* as any other kind of household electrical device is also considered correct.

which got a large positive transfer (+25%) and the *lamp* class which got a large negative one (-10%) when training with 10 examples per class for all classes, as described in the paper.

Fig. 1 shows the results on the *orchid* class. Fig. 1a shows the classification accuracy as the number of examples of *orchid* given to the algorithm increases from 5 to 500. We see that all the models do about the same, with the tree-based models doing slightly better. This is probably because the given tree structure is quite accurate for the *flower* superclass which includes *orchid*. Learning the tree does not lead to much change. Fig. 1b shows the classification accuracy when classifying an example as any other kind of flower is also considered correct. Here we see that the tree-based models perform significantly better, showing useful positive transfer. This is expected because the tree-based prior encourages the classification parameters for all kinds of flowers to be close together.

Fig. 2 shows the results on the *lamp* class. Fig. 2a shows the classification accuracy as the number of examples of *lamp* is increased. We see that the fixed tree model does worse than the other two. This is probably because *lamp* falls in the superclass *household electrical devices* which includes visually dissimilar objects such as keyboards and clocks. The learned tree model however does better. It learns to not group *lamp* along with these dissimilar classes and almost matches the baseline's performance. Fig. 2b shows the accuracy when classifying *lamp* as any other household electrical device is also considered correct. In this case, the fixed tree model does much better. This is probably because the prior used by the fixed tree favours this loss function. This shows that the model is able to enforce relationships that are implied by the tree structure.