[Reviews · NeurIPS 2013]

Submitted by Assigned_Reviewer_4

The paper presents a multi-task learning approach based on hierarchical Bayesian priors. The priors are combined with a deep NN resulting in a discriminative learning process which encourages sharing information among tasks taking into account different levels of relatedness.

The paper is written well and easy to follow. The topic of multi-task learning is a very popular and important topic and the sub issue of considering different level of relatedness has also gained much attention lately. I liked the proposed integration with NN and the good experimental results.

I feel the paper does not give enough background on related approaches. There are many related papers, but specifically I feel a discussion on similar hierarchical Bayesian approaches is lacking:

Bayesian Multitask Learning with Latent Hierarchies, H. Daume III

where a similar prior is seemed to be used for what they denote the domain adaptation approach.

Further related work is:

Multi-task learning for classification with Dirichlet process priors, Xue et al. 2007

and more papers presenting discriminative approaches dealing with different levels of relatedness:

Learning with whom to share in multi-task feature learning, kang et al.

Hierarchical regularization cascade for joint learning, zweig et al.

Tree-guided group lasso for multi-task regression with structured sparsity. Kim et al.


I also have a few questions regarding the algorithm for discovering the hierarchical structure:

- The hierarchy discovery is described in the context of the CRP prior but then it is said that solving the inference problem directly is too complex and thus an approximation is proposed. How does this approximation relate to the original prior? I couldn't find the explicit use of the prior (over the generation of a new super class) in the presented approach.

- Unless I missed something, it seems that the current presentation and experiments deals only with a two level hierarchy, super class and original classes. Can the approximation method be at all extended to more levels?

- The method requires a validation set in order to decide on the best model at each step of the hierarchy construction algorithm. Can we assume a validation set in small sample scenarios (which are really the motivating scenario for multi-task learning)?
I saw in the supplementary material this issue is addressed, but didnt understand if the validation set discussed there is the validation set for the hyper-parameters or the hierarchy discovery or both.

- How is the convergence of the hierarchical structure measured?

- The algorithm assumes an initial good guess. What can be done when such a guess is not available or too costly? would a random initialization work well? It would be nice to see an experiment with a random initialization


Generally I liked the experimental section and found it sufficient. My only concern is that I would have preferred to see the experiment on small sample of only a single class done on more examples, then only the dolphin class.
Summary: The approach is nice and shows good experimental results. I would like the paper to better present its novelty in the context of the rich literature and address the issues I raised regarding the structure discovery algorithm presented.


Read the rebuttal, the authors addressed my main concerns and will update the final version according to the points raised in the reviews.

Submitted by Assigned_Reviewer_5

This paper presents an algorithm for deep learning architectures to jointly learn to classify images and a hierarchy of classes, such that "poor classes" (those that have less training examples) can benefit from similar "rich classes" (those that have more training examples). Experiments on CIFAR-100 and MIR-Flikr datasets show improvements.

Other papers have been proposed to learn hierarchies (and a few are mentionned in the paper indeed, but were not deemed applicable). I was thinking of the label embedding trees (Bengio et al, NIPS2010) as it seems it could have been applied in this context, if I understood both correctly. Another related approach, to my mind, is the hierarchical softmax, which is very useful with a large number of classes.

I was wondering about eq (1) as it seems to assume that w and beta are two independent sets of parameters, while a deep learning architecture would learn them jointly, thus making w such that beta_i would be as independent as possible with respect to each other. Clearly, the representation vector (in R^D) has been learned to take into account all classes jointly, and hence in that space, I'm expecting similar classes to be nearby.

Regarding the algorithm to learn the tree, it seems to be limited to 2-level trees, unless I missed something. Is that a hard limitation? I was thinking that with large number of classes, a deeper hierarchy would make more sense.

Regarding training complexity, could you provide some idea about how slower the proposed algorithm is, with respect to the number of classes or super-classes for instance?

Regarding results, clearly the "poor classes" seem to benefit from the "rich classes". But I'm expecting the performance of the "rich classes" to sometimes drop because they have shared where they didn't need to. This could be bad since often the "rich classes" are those popular objects which you don't want to miss. Could you verify if this is the case? You said for instance that 30% of the classes had lower performance. Were they the rich classes?
Summary: A new algorithm is proposed to learn hierarchies of classes and is expected to be good when the class distribution is not uniform.
The algorithm is sound but limited to 2-level hierarchies (I think). Experimental results are good, but not compared to any alternative.

Submitted by Assigned_Reviewer_6

The paper is written very clearly and easy to understand. The details of the experiments and hyper parameters are explained very clearly too.

The model described in the paper assumes that the tree over class labels is limited to 2 layers only, such that any class label is affected by its immediate parent only. As far as I can see, in terms of deriving the updates for the tree and class labels, the only difference from ref. 19 is formulating the prior of a class label (\beta_k) by a gaussian with mean at its parent, which results in the same closed form solution for \beta_k as given in eq. 5. Of course, additionally the convolutional net is updated using SGD updates.

The model is evaluated on two datasets, cifar 100 and MIR Flickr. I think there are two aspects of improvement in this model. The first one comes from sharing examples between similar classes and the second one comes from transferring knowledge to a class with few number of examples. Experimental evaluations in sections 3.1 and 3.2 investigate these cases respectively. In the first case, one can see that by introducing the tree prior, the classification performance of around 70% of classes are improved which results in overall improvements ranging from 6% to < 1% depending on number of training samples per class. The more interesting second experiment is done on only a single target class (dolphin) where the number of training samples is varied between 5 tp 500. Using the tree prior, the classification performance of dolphin class is improved by 3% across a wide range of training samples per class. This example shows that with the given model, it is actually possible to transfer knowledge from classes with many samples to classes with few samples.

On line 86, the paper claims that they show that learning features on pixels is important for being able to learning from few examples. However, I do not see any explanation or experimental validation supporting this point.

Also, it seems to me that the particular contribution of this paper over ref. 19 seems to be limited to using a convolutional net introduced in ref.11 instead of a battery of engineered feature extractors.

Summary: The paper presents a tree based classification approach that motivates learning the tree structure in order to transfer information between similar classes and especially improve classification performance of classes with very few samples from similar classes. The setup is very similar to a previous paper with the exception that current paper proposes a model that is trained end-to-end on pixels using a popular recent convolutional net model.
Author Feedback

Author rebuttal: We thank the reviewers for their insightful comments and suggestions.

In the final draft we will add a more detailed discussion of related approaches and highlight the key differences from our approach, most notable of which is the integration of a hierarchical Bayesian prior with a discriminatively trained neural net, focusing on transfer learning from few examples, which is something that is missing in many of the deep learning approaches.

The model can be extended beyond two-level hierarchies. The key idea is that each parent node defines a common shared prior for its children. This can be applied to construct trees of any depth. However, for the considered datasets we do not believe that adding additional levels of hierarchy will substantially improve our results.

Reviewer 4:

The CRP prior is used, along with the likelihood, for scoring different trees when searching over tree structures. The loss function in Line 210 evaluated on a validation set was used to choose the best model. The CRP prior affects the P(z) term in this loss function. It encourages the model to share parent nodes rather than create new ones. The same validation set was used for hyperparameter optimization.

It is true that this is more challenging when the number of labeled examples is small for some classes. However, we found that in order to discover meaningful hierarchies, it was very important to use a validation set, even if it contained only a handful of examples.

If the position of any class in the tree did not change during a full pass through all the classes, the hierarchy discovery was said to have converged.

If we do not have a good initial hierarchy, we can use hierarchical clustering over the top-level features in deep neural nets, which will provide us with a good initial guess. We could also start with a randomly initialized tree. We will add the results obtained when starting from a random initialization in the final draft. But we should note that for many of the considered problems, we almost always have a reasonable hierarchy to start with (in our experiments these are semantic hierarchies that can be easily derived from the Wordnet).

Similar to the ‘dolphin’ class, we obtained improvements on many other classes as well. We will include those results in the final draft of the paper.

Reviewer 5:

Both label embedding trees and hierarchical softmax are interesting alternative approaches. However, unlike these, our model has an important property that it only acts as a prior. Therefore, as the number of examples for a particular class increases, the effect of the prior will decrease, and it is no longer forced to share. This allows our model to smoothly deal with data sets which contain both frequent and rare classes.

The time complexity of making a single hierarchy search pass is O(#superclasses * #classes). This search can be parallelized to reduce the time to O(#classes) per pass. The number of passes is determined by when the stopping condition is met (see Reviewer_4). We typically need to make 3-5 passes.

We found that the “rich” classes do not suffer much as shown in Figure 6(b). We suspect that this is because for rich classes, the effect of the prior diminishes. In other words, \beta_{richclass} can afford to ignore the prior and be far from its parent if that helps the likelihood term in Eq(4). The classes which see a drop in performance are usually those which were put in the wrong superclass.

Reviewer 6:

Similar to the ‘dolphin’ class, we obtained improvements on many other classes as well. We will include those results in the final draft of the paper (see response to reviewers 4 and 5).

The prior in ref. 19 uses a sum of weights along the path from the root to the leaf, which is different from the prior used in our model. We did experiments using the prior from ref 19 and found that it did not work as well, at least on our data sets. Another important difference is that we propose a method for jointly learning low-level features, high-level features and a class hierarchy, whereas ref 19 only considers learning the hierarchy.